# Leaving Past Adversities Behind: Gratitude Intervention Compensates for the Undesirable Effects of Past Time Perspectives on Negative Affect

**DOI:** 10.3390/ijerph191912964

**Published:** 2022-10-10

**Authors:** Bozena Burzynska-Tatjewska, Maciej Stolarski

**Affiliations:** 1Faculty of Psychology, SWPS University of Social Sciences and Humanities, 03-815 Warsaw, Poland; 2Faculty of Psychology, University of Warsaw, 00-183 Warsaw, Poland

**Keywords:** time perspective, gratitude, savoring the moment, positive affect, negative affect

## Abstract

Both gratitude and savoring the moment are considered to be well-established “well-being boosters” (WBBs). Each of them has a salient temporal reference: Gratitude is past-related, whereas savoring the moment refers to the present. The temporal match–mismatch model posits that time perspectives (TPs) moderate the effects of WBBs on well-being if they refer to the same temporal frame (e.g., gratitude and Past-Positive). Our study tested whether TPs moderate the effects of two positive interventions on positive affect (PA) and negative affect (NA). The participants (N = 174 individuals, 73% women) completed measures of TPs, PA, and NA, followed by a brief intervention of gratitude (N = 86) or savoring (N = 88). Subsequently, they completed the PA/NA measures again. Both interventions significantly decreased NA but did not foster PA. The magnitude of the shift in NA in the gratitude condition depended on the levels of past TPs: Individuals high in Past-Negative and low in Past-Positive displayed greater baseline NA than their counterparts; however, the difference was leveled after the gratitude intervention. No interaction effects between the present TPs and the savoring intervention were observed. The results suggest that practicing gratitude may diminish the undesirable consequences of negative views of the past.

## 1. Introduction

Since the discipline of positive psychology first emerged, it has grown rapidly by examining such constructs as virtues and character strengths [1]. In addition, positive psychologists have started to look for ways to improve positive emotions, lessen distress, and generally foster subjective well-being. Examples include optimistic interventions [2] and empathy-based interventions [3].

Recently, Burzynska and Stolarski [4], building on the framework of time perspective (TP) theory [5,6], noted that some established well-being boosters (WBBs) have built-in temporal references. They focused on three widely cited constructs from within the scope of positive psychology: gratitude [7], savoring the moment [8], and prioritizing positivity [9], emphasizing their temporal nature. Burzynska and Stolarski highlighted the connections between the past and gratitude, between the present and savoring the moment, and between the future and prioritizing positivity. They also proposed four conceptual models for illustrating the possible dynamics between each of the three WBBs and those TP dimensions that refer to respective temporal horizons. Their predictions were subsequently tested in both cross-sectional and longitudinal studies [10,11], providing some support for the so-called ”trait-behavior model”, which assumes that individual tendencies to focus on particular TPs may foster the probability of making use of temporally coherent WBBs (e.g., Past-Positive and gratitude) [11]. Two other conceptual models, i.e., the accumulation model, which predicts the opposite direction of the relationship (WBBs influencing well-being via fostering adaptive change in TPs), and the feedback loop model, which assumes reciprocal effects of TPs and WBBs on each other, were not supported in those earlier studies [10,11]. Interestingly, the empirical test of another model, which assumes interactions between TPs and temporally respective WBBs (referred to as the match–mismatch model), led to surprising conclusions [10]. The results supported the hypothesis about the interactions between TPs and WBBs that refer to the very same horizon. However, contrary to the authors’ expectations, significant moderation effects emerged for the “negative” TP dimensions (e.g., Past-Negative), not for the “positive” ones (e.g., Past-Positive). The results suggested that, unlike Burzynska and Stolarski [4] initially predicted, “positive” TP dimensions do not enhance the effectiveness of temporally bounded WBBs (e.g., Past-Positive did not foster the effects of gratitude on well-being). On the other hand, the tendencies to use particular WBBs interacted with the “negative” TPs, which were related to the same temporal horizon. The authors concluded that the tendency to make use of particular WBBs may compensate for the undesirable consequences of those TPs that are associated with respective temporal horizons. For instance, gratitude may diminish the negative consequences of Past-Negative [10]. In that earlier study, the authors applied a cross-sectional design. Taking into account the limitations of that approach, in the present study we attempted to re-test the match–mismatch model using an experimental methodology. 

Given the complete lack of conclusive results for prioritizing positivity in the previous studies [10,11] that tested the models proposed by Burzynska and Stolarski [4], in the present research, we focused on two of the three WBBs with marked temporal anchoring: gratitude and savoring the moment. We aimed to provide an empirical analysis of the interplay between respective TPs, gratitude, savoring the moment, and affective states. Therefore, our major goal was to test whether particular TPs moderate the effects of gratitude intervention and savoring the moment intervention on positive affect and negative affect, thereby providing a direct experimental test of the match–mismatch model [4]. Moreover, determiningwhether the effectiveness of the positive interventions depends on individual TP profiles may prove useful for positive psychology practitioners who aim to provide their clients with customized interventions, taking into account their psychological profiles. 

### 1.1. Conceptual Background for the Current Research

Building upon the classic concept of TP that was introduced by Lewin [12], Zimbardo and Boyd defined the phenomenon as “the often non-conscious personal attitude that each of us holds towards time and the process whereby the continual flow of existence is bundled into time categories that help to give order, coherence, and meaning to our lives” ([13], p. 51). Despite the dynamic, processual nature of the temporal framing of current experiences, individuals develop relatively stable tendencies to focus on particular temporal horizons (past, present, and future), usually combined with specific attitudes toward each of them. Based on their conceptual and psychometric analyses [5], Zimbardo and Boyd distinguished five basic dimensions of TP: Past-Positive (referring to a pleasant, warm perception of the past), Past-Negative (depicted in a negative, aversive view of the past, filled with traumas and regrets), Present-Hedonistic (describing an orientation toward immediate pleasure, risk-taking tendencies, and impulsiveness), Present-Fatalistic (describing an orientation of hopelessness and helplessness), and Future (indicating a broad outlook on the future, consideration of future consequences, and focus on distant goals). Later research [14] further distinguished between two forms of the prospective temporal focus: Future-Positive (emphasis on chances and personal goals, but also taking into account the long-term consequences of present actions) and Future-Negative (fear of the future and attention to threats). In the present study, we adopted the six-factor conceptualization of TP. 

TPs have often been discussed and studied within the scope of positive psychology and happiness studies (e.g., [15]). The associations between particular TPs and both cognitive and affective components of well-being are well documented. For instance, Zhang et al. [16] reported consistent links between particular TP dimensions and such features as life satisfaction, happiness, or positive/negative affect, with the most pronounced effects observed for Past-Negative. Nevertheless, these studies applied cross-sectional research designs; therefore, little was known about both the actual causal effects of TPs on well-being and the role of temporal perspectives in shaping the magnitude of psychological interventions aimed at alter emotional states and fostering well-being. The moderating role of TPs on the effectiveness of positive interventions, including gratitude and savoring interventions, also remains unstudied.

According to theory and research, gratitude has favorable effects on well-being indicators [17,18]. In the light of recent studies, gratitude appears to be positively correlated with positive affect, happiness, and life satisfaction and negatively correlated with negative affect and depressive symptoms [19,20]. The gratitude list, in which subjects identify three to five items they were grateful for during the day, is one of the treatments that has been most thoroughly applied and studied [21,22]. Research results confirmed that this easy and short activity can promote increased positive affect (PA) [23] and life satisfaction [24], as well as decrease negative affect (NA) [25,26]. Although the results of interventions promoting gratitude are promising [25], little is known about personality features that may impact the occurrence and magnitude of these effects.

Savoring the moment occurs when one extends or enhances the good sensations connected to present events via certain thoughts or activities [27]. Bryant and Veroff [8] discussed ten strategies facilitating or enhancing savoring, including interpersonal (e.g., talking about the event with others) and cognitive (e.g., intentionally storing details of the experience for later recall) techniques. Empirical research provided evidence that savoring interventions result in an increase in mental well-being (e.g., [28,29]). For instance, participants of a present-focused savoring intervention expressed more positive emotions after the intervention, compared with the control group [30]. Another study showed that individuals who participated in a savoring-the-moment intervention reported significantly reduced levels of depression and negative affect, but there was no difference in positive affect during the period of two weeks [28]. Finally, according to a meta-analysis, savoring interventions generally increase happiness and positive affect [31]. Again, factors that may impact the magnitude of the effect of savoring remain unknown.

### 1.2. The Present Study

The present study examined whether TPs moderate the emotional consequences of brief gratitude and savoring interventions. To the best of our knowledge, no research has tested whether people feel more positive (or less negative) immediately after practicing gratitude or savoring, depending on their habitual tendencies to focus on particular time horizons.

We decided to conduct the study online. Online studies carried out on adult populations provided evidence that online positive interventions remain effective [32,33]. For instance, Banos et al. [34] reported successful outcomes of a self-guided Internet intervention to promote pleasant emotions and strengthen psychological resources. In a randomized, placebo-controlled study conducted online, Gander et al. [35] revealed the effects of nine strengths-based positive interventions on happiness and sadness. 

Previous studies on TPs, WBBs, and well-being supported certain predictions with respect to gratitude and savoring, but not for prioritizing positivity [10,11]. It seems that the latter method for enhancing happiness remains unrelated to positive future thinking; therefore, in the current study, we focused on gratitude and savoring the present. 

Based on the conceptual framework provided by Burzynska and Stolarski [4], as well as the results of the seminal empirical test of the match–mismatch model [10], we hypothesized that the effectiveness of the gratitude intervention (operationalized as a gain in PA and a decrease in NA) is moderated by Past-Positive (H1) and Past-Negative (H2) TPs. More positive (and less negative) views of the past could provide more cognitively available happy or pleasurable events from the past; therefore, the positive effects of the gratitude practice in individuals with higher levels of Past-Positive (and lower Past-Negative) may prove more pronounced. On the other hand, people with less adaptive TP profiles (low Past-Positive, high Past-Negative), may benefit more from gratitude practice—becoming aware of the things that they are grateful for may allow them to compensate for more negative baseline affects that are characteristic of individuals with a similar perspective of the past [6]. Moreover, we expected that analogical effects occur for the savoring intervention and both Present-Hedonistic (H3) and Present-Fatalistic (H4). Given that the original conceptual analysis by Burzynska and Stolarski [4] hypothesized a positive interaction (e.g., enhancement of the effects of gratitude on well-being or affective balance by Past-Positive), whereas the cross-sectional research provided evidence for compensation effects (e.g., the effects of gratitude were more pronounced among individuals with high Past-Negative; see [10]), we did not decide to make any predictions with respect to the direction of the moderation effects. Finally, assuming the temporal specificity of these effects, we expected that the hypothesized interactions are time horizon-specific (i.e., that the interactions occur between WBBs and TPs that share the same temporal reference—e.g., past: gratitude × Past-Positive—but not for the WBB-TP dyads with a distinct temporal reference—e.g., gratitude (past-oriented) and Present-Hedonistic).

## 2. Materials and Methods

### 2.1. Participants and Procedures

Polish adults (N = 174; 73% women), aged 18–71 years (M = 35,32; SD = 10.29) participated in the study. The data were collected in an online study via the Qualtrics platform. The participants were recruited through social media platforms. After providing informed consent, participants completed two questionnaires, measuring TPs and positive and negative affect, followed by one of the two positive interventions—gratitude (N = 86) or savoring the moment (N = 88) condition. Subsequently, the measure of positive and negative affect was applied once again. To create equivalent groups, participants were randomly assigned to one of the two experimental conditions. The entire procedure was anonymous. Demographic data were collected, and included questions about age, gender, and education. The study took approximately 30 min. It was confidential and the participants received no remuneration. In order to avoid missing data, each question on the online form required a response. The study met the ethical standards of the Declaration of Helsinki and was approved by the Ethics Committee at the Institute of Psychology of SWPS University of Social Sciences and Humanities, Warsaw, Poland. All participants received feedback on the study’s overall findings. 

### 2.2. Measures

The Zimbardo Time Perspective Inventory was used to measure individual differences in TPs (ZTPI; [5]). Five TPs—Past-Negative, Present-Hedonistic, Past-Positive, Present-Fatalistic, and Future—were measured according to the original version of the inventory. The questionnaire was later revised by Carelli et al. [14], who supplemented it with Future-Negative items and slightly revised the original Future scale to give it an unequivocally positive valence. In the questionnaire, respondents ranked their agreement with each statement on a scale ranging from 1 (very uncharacteristic) to 5 (very characteristic). The Polish adaptation’s adequate reliability (Cronbach’s alphas for Past-Negative, Past-Positive, Present-Hedonistic, Present-Fatalistic, Future-Positive, and Future-Negative, respectively, were 0.86, 0.70, 0.82, 0.75, 0.76, and 0.71) and construct validity were demonstrated by Jochemczyk et al. [36].

A Positive and Negative Affect Scale was used to measure momentary levels of positive affect and negative affect (PANAS; [37]). This 20-item self-reporting measure of current mood was bidimensional. Responses were rated on a five-point Likert scale, ranging from 1 (very slightly or not at all) to 5 (extremely). Watson et al. [37] provided evidence of the questionnaire’s good internal consistency (Cronbach’s alpha amounted to 0.87 for PA and 0.85 for NA).

### 2.3. Gratitude and Savoring the Moment Manipulations 

Both instructions were appropriately modified in order to be as similar as possible in terms of length and structure. The instructions differed in their content, aiming to elicit a specific psychological state: either gratitude or savoring the moment.

For the gratitude manipulation, we used the instructions proposed by Emmons and McCullough [21]. The adjusted manipulation was as follows: “There are many things in our lives, both large and small, that we might be grateful about. Think back over the day and write down on the lines below all that you are grateful for today. It could be something small, like the smell of coffee or a free afternoon, or something more serious, like a pleasant meeting or success. Write down below up to five things you are grateful for. After writing each one, close your eyes and imagine it for about 20 s”.

The savoring intervention used an adjusted version of the manipulation proposed by Smith and Hanni [38]. The adapted manipulation was as follows: “First, think of something good/pleasant that is happening right now. It could be something small, like the smell of coffee, a free afternoon, or something more serious, like the anticipation of a meeting. Ask yourself, what is it about the experience that you find so enjoyable? Next, notice the positive feelings that occur when you think about the experience (e.g., amusement, interest, excitement, contentment). Finally, take a moment to appreciate the experience. Think about how special it is and how good it is that it happens to you. Write below what event/thing you recalled and what emotions came up in you”. The original intervention was used for all three kinds of savoring: past, present, and future. 

### 2.4. Statistical Analyses

SPSS 27 (SPSS Inc., Chicago, IL, USA) for Windows was used to conduct all of the statistical analyses. Pearson’s correlations were used for the preliminary analyses. Dependent *t*-tests were used for the analyses of the effectiveness of the positive interventions. The main hypotheses were tested using repeated measurement ANCOVAs.

## 3. Results

### 3.1. Preliminary Analyses

In order to provide a general overview of the interrelationships between the measured constructs in Table 1, we provided descriptive statistics and the matrix of correlations between variables measured in the entire sample (N = 174). The pattern of associations between TPs and PANAS subscales was consistent with previous studies. Additionally, we performed analogical analyses separately for participants from each experimental condition (see Appendix A, Table A1). Both the descriptive statistics and the pattern of intercorrelations were very similar for the two groups of participants. 

### 3.2. Effectiveness of Manipulation

In order to test whether the experimental manipulations were effective, we conducted a series of dependent *t*-tests. In the gratitude condition, we observed a small, albeit statistically non-significant, improvement in PA (*t*(85)= −1.31, *p* = 0.194, Hedges’ g = −0.14), and a robust, statistically significant decrease in NA (*t*(85) = 5.68, *p* < 0.001, Hedges’ g = 0.61). 

In the savoring condition, a similar pattern of results emerged, with a lack of significant change in PA (*t*(85)= −1.38, *p* = 0.172, Hedges’ g = −0.15) and a marked, significant decrease in NA (*t*(85)= 6.81, *p* < 0.001, Hedges’ g = 0.72) (see Appendix A, Table A1 for the means and standard deviations in both conditions). Thus, both interventions proved to be effective, however, their effectiveness was only with respect to NA. 

### 3.3. Moderation Analysis;

Next, we attempted to test our major predictions regarding the moderating effect of TPs on the effectiveness of the positive interventions. Given that preliminary analyses showed that both interventions led solely to a decrease in NA but not in PA, in the further analyses we focused on the former variable. 

In order to test whether TPs moderated the effects of the positive interventions, we computed a series of ANCOVAs with measurement time (pre- and post- intervention NA) introduced as a within-person factor, while TP dimensions were added as covariates (a single TP dimension was introduced in each analysis). We decided to avoid transforming continuous variables (i.e., TPs) into discrete variables at the stage of statistical treatment in order to avoid the loss of a significant portion of variance, which remains inevitable when transforming continuous variables into nominal scales. Nevertheless, we did use cut-offs for the graphical illustration of the interaction effects (see Figure 1 and Figure 2). Given that the temporal match–mismatch model predicts the occurrence of the interaction effects only in a situation of shared temporal reference between a TP dimension and WBB, we focused on four potential interaction effects: gratitude × Past-Positive, gratitude × Past-Negative, savoring × Present-Hedonistic, and savoring × Present-Fatalistic.

In the first of these analyses, we observed a lack of a statistically significant effect of Past-Positive on NA, F(1,84) = 0.47, *p* = 0.49, η^2^ = 0.006. The effect of intervention remained significant, F(1,84) = 10.078, *p* = 0.002, η^2^ = 0.11, whereas the interaction effect amounted to F(1,84) = 4.61, *p* = 0.035, η^2^ = 0.05. The graphical illustration of the effect is provided in Figure 1. It shows that the magnitude of the decrease in NA after the gratitude intervention was greater among individuals displaying lower levels of Past-Positive perspective. 

In the second analysis, we observed a strong, significant effect of Past-Negative on NA, F(1,84) = 24.32, *p* < 0.001, η^2^ = 0.22. The effect of intervention was no longer significant in the model, F(1,84) = 0.12, *p* = 0.73, η^2^ = 0.001, whereas the interaction effect amounted to F(1,84) = 3.68, *p* = 0.058, η^2^ = 0.04 (see Figure 2 for a graphical representation of the results). This showed that the magnitude of the gratitude intervention-related shift in NA was greater among individuals who were characterized by the negative views of their past. 

Subsequently, we conducted an analogical analysis for the present TPs and savoring. However, none of the models provided evidence for the hypothesized interactions between these dimensions, with F(1,86) = 1.60, *p* = 0.21, η^2^ = 0.02 for the interaction between savoring and Present-Hedonistic and F(1,84) = 1.18, *p* = 0.28, η^2^ = 0.01, for the interaction between savoring and Present-Fatalistic.

Given that the lack of a significant main effect of the intervention does not rule out the occurrence of interaction effects, we have additionally carried out analogical tests for PA; however, none of the hypothesized interactionss proved to be significant.

## 4. Discussion

The current study investigated whether individual differences in TPs can impact the emotional effects of brief interventions that promote gratitude and savoring the moment, in a way predicted by the temporal match–mismatch model [4]. While neither intervention significantly enhanced the level of PA, they both resulted in a significant decrease in NA, providing further evidence of their effectiveness in the reduction of negative emotions. The results seemed in-line with earlier research that showed that certain savoring practices may be more effective in decreasing unpleasant moods than in boosting happy feelings [28]. Other studies also suggested that gratitude interventions are successful in the immediate reduction of the negative affect, but not necessarily in the immediate promotion of the positive affect (e.g., [25,39]). The lack of an increase in positive affect might also be explained by the length of the intervention: It seems possible that a long-term, persistent effort is necessary to experience gratitude and a savoring-related increase in PA. Therefore, our findings contribute to the growing body of research that suggests that gratitude practice decreases negative affect, lowers stress levels, and might play a buffering role against the harmful psychological consequences of stressful life events [39,40]. 

Furthermore, as expected, this study revealed that the Past-Negative perspective was positively correlated with NA, and negatively correlated with PA in both pre- and post-intervention measurement. Present-Fatalistic was positively associated with NA, pre- and post-test. These and other associations between TPs and the two affective dimensions were slightly weaker, albeit consistent with previous studies [16,41].

Our main goal was to examine the moderating role of TPs in shaping the affective response to the two brief positive interventions mentioned above. Based on the conceptual analysis [4] and the results of the seminal tests of the match–mismatch model [10], we sought empirical evidence for the interplay between particular TPs and WBBs that shared the same temporal reference. However, given that the initial theoretical considerations predicted that positive TPs may enhance the effects of WBBs that referred to the same time frame [4], while empirical evidence suggested, instead, that WBBs may allow compensation for the undesirable consequences of the negative TP dimensions [10], we decided not to make any predictions regarding the direction of the interaction. We predicted that Past-Positive (H1) and Past-Negative (H2) TPs moderate the effectiveness of the gratitude intervention. Additionally, we anticipated similar effects for the savoring intervention and both Present-Hedonistic (H3) and Present-Fatalistic (H4). 

The present results showed that individuals scoring high on Past-Negative and low on Past-Positive TP dimensions experienced greater decreases in the levels of NA as a result of the gratitude intervention. Therefore, Hypotheses H1 and H2 were supported; however, given that the effect obtained for Past-Negative TP did not reach the *p* = 0.05 significance threshold, the result should be treated with caution. The results are generally in-line with the empirical findings reported by Burzynska-Tatjewska et al. [10], but they are opposite to the predictions of the conceptual match–mismatch model [4], which predicted synergy effects between particular TPs and temporally related happiness boosters. It seems that both cross-sectional [10] and experimental designs provide evidence for a compensatory nature of the analyzed interplay: gratitude, operationalized both as a trait and as a transient, experimentally elicited psychological state, allows for a (partial) reduction of the undesirable emotional consequences of the elevated Past-Negative and low Past-Positive temporal perspectives. In light of these data, practicing gratitude seems to be an effective way of coping with elevated negative emotions experienced by individuals who are characterized by a rather negative or aversive view of the past, filled with traumas and regrets (i.e., high Past-Negative), as well as those who did not develop a positive, warm, and sentimental view of their personal past (low Past-Positive). 

One may wonder what exact psychological mechanism underlies the obtained interaction effects. In our opinion, the roots of the compensation effect may be twofold. First, positive, happy, self-enhancing past experiences may become a source of vital psychological resources [13], but for people who did not develop the habit to revisit the positive past memories, they remain inaccessible, like a long-forgotten buried treasure. Gratitude practice may provide a way to retrieve the treasure out of the vault of the memory and transform it into tangible benefits [42]. On the other hand, a regular gratitude practice may make the same memories much more familial and less powerful in terms of their impact on emotion, due to the ubiquitous mechanism of habituation [43]. Such an explanation may explain the effects presented in Figure 1, which shows the individuals with low levels of Past-Positive not only caught up with their counterparts, but even overtook them in their shift towards extinguishing negative emotions. While participants with higher levels of Past-Positive are probably well accustomed to their happy memories, those with low Past-Positive scores may have rediscovered them during the gratitude exercise. As a result, the desirable decrease in negative emotions was greater among the latter than among the former. 

The other explanation refers to the second interaction effect (i.e., the one between Past-Negative and gratitude (see Figure 2). In general, improvements in mood often depend on basic levels: the lower the mood before an intervention, the greater the improvement afterwards, in line with the ceiling effect [44]. An attentive reader noticed, plausibly, that the levels of NA among individuals with low levels of Past-Negative were already very low prior to the intervention (see Figure 2). Accordingly, in their case, there was only very little “space” for improvement (at least at the level of measurement). On the other hand, the levels of NA among individuals with high Past-Negative were relatively high and, as a result, the “space” for an improvement was clearly greater, which allowed for a greater decrease in negative emotions.

Unlike the cross-sectional study that found some evidence for the interaction between Present-Fatalistic and savoring [10], we found no evidence for the match–mismatch effects for the present TP dimensions and the savoring intervention. Therefore, hypotheses H3 and H4 were not supported. This negative result may stem from the problematic nature of the present TP dimensions that are included in Zimbardo and Boyd’s [5] model. Content analyses conducted by Stolarski, Fieualine and Zimbardo [6] suggested that neither Present-Fatalistic nor Present-Hedonistic reflected an actual focus on the present. While the former indicates a sort of temporal helplessness and an external locus of control over temporal foci, the latter depicts a focus on immediate consequences and striving for pleasure; therefore, it captures a short-term future orientation. The authors emphasize the need for broadening the basic TP universe with another form of present orientation and point to such recently introduced constructs as the Eudaimonistic Present [45] or Carpe Diem orientation [46]. These novel temporal concepts have been developed with a direct reference to such phenomena as mindfulness or flow; hence, their use of the “present” label with respect to these temporal perspectives seems far more justified. Future studies should attempt to revisit the match–mismatch model using these novel temporal dimensions. 

It is worth noting that in our separate analysis, which is not reported in the present paper, we tested for interactions between TPs and WBBs that refer to different time horizons. For instance, we analyzed whether past TPs moderate the effects of savoring on PA and NA, or whether present TPs interact with gratitude intervention. None of these interaction terms proved to be significant, which may be interpreted as a robusticity check for the present findings, and which may provide evidence for temporal specificity of the interplay between TPs and WBBs. As predicted in the match–mismatch model, the interplay between TPs and WBBs occurs only when they share their temporal reference.

## 5. Limitations

This study has limitations that should be addressed in future research. In the absence of a control group, we were unable to completely eliminate the possibility that the changes in the outcome variables were caused by outside factors. Randomized control studies are required to examine the interventions’ efficacy in a more accurate way. Therefore, future studies should include both experimental and control groups. In addition, the sample size was limited, and females were overrepresented; therefore, a direct replication of the study would be highly desirable. 

Moreover, this study had all of the limitations associated with online data collection [47], including the fact that the sample was limited to the population of social media users, which obviously limits the generalizability of the present results. Nevertheless, given that the study applied an experimental design, the lack of representativeness of the studied sample should not influence the results to a considerable degree. Our study focused mainly on the change in affective states, rather than on any baseline sample characteristics that could differ between users and non-users of social media. 

Another limitation of this study is that it relied on self-reported measures, which were susceptible to self-presentation and response bias. Furthermore, certain limitations of mood measurement (e.g., the ceiling effect mentioned above) might have influenced the present findings. Moreover, we did not include the third temporal WBB that was conceptually analyzed by Burzynska and Stolarski [4]: prioritizing positivity. The decision to abandon this intervention was based on the results of previous studies; however, given that the studies only measured self-reported prioritizing positivity, it is still possible that this form of positive intervention could interact with TP dimensions in shaping affective responses. 

Finally, in the present paper, we analyzed solely transient mood responses, whereas previous studies in this area [10, 11] applied aggregate measures of subjective well-being. Future studies should replicate the present findings, using prolonged interventions (e.g., completing a gratitude diary over a month) and verifying their impact on general well-being levels.

## 6. Conclusions

In conclusion, this study provided evidence of the interplay between Past-Negative and Past-Positive and gratitude intervention in shaping a momentary negative affect. Individuals with high levels of Past-Negative and low levels of Past-Positive TP might benefit more from practicing gratitude than their counterparts. Additional studies are needed to further examine the role of individual differences in temporal framing in shaping the effectiveness of positive interventions, with a clear temporal reference, such as gratitude or savoring. While more research is called for, gratitude interventions appear to be an affordable/cost-effective method for reducing the unpleasant emotional consequences of negative views of the past.

## Figures and Tables

**Figure 1 ijerph-19-12964-f001:**
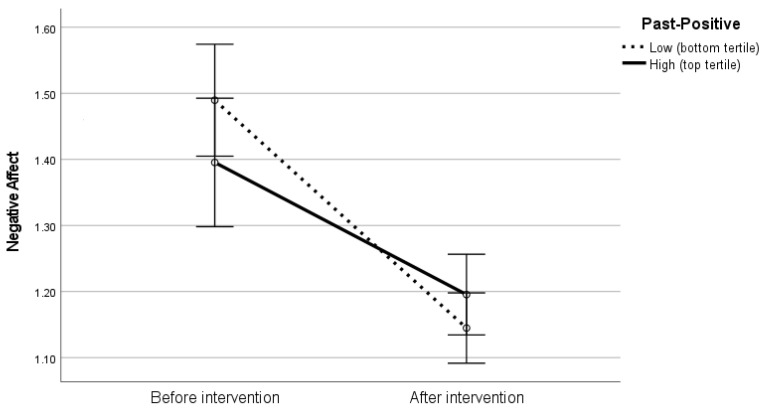
Interaction between gratitude intervention and Past-Positive perspective in predicting negative affect.

**Figure 2 ijerph-19-12964-f002:**
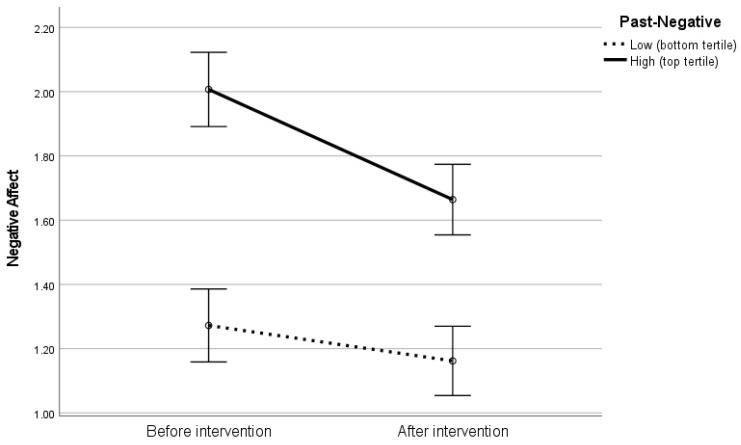
Interaction between gratitude intervention and Past-Negative time perspective in predicting negative affect.

**Table 1 ijerph-19-12964-t001:** Means, standard deviations, Cronbach’s alphas, and intercorrelations between variables included in the study (N = 174).

	Variable	M	SD	α	1	2	3	4	5	6	7	8	9
1	Positive affect (1)	2.75	0.77	0.87									
2	Negative affect (1)	1.50	0.60	0.86	−0.28 **								
3	Positive affect (2)	2.82	0.88	0.91	0.84 **	−0.25 **							
4	Negative affect (2)	1.27	0.50	0.90	−0.21 **	0.81 **	−0.28 **						
5	Past-Negative	2.91	0.80	0.85	−0.29 **	0.44 **	−0.29 **	0.38 **					
6	Past-Positive	3.57	0.64	0.74	0.16 *	−0.14	0.11	−0.05	−0.24 **				
7	Present-Hedonistic	3.25	0.57	0.82	0.16 *	0.09	0.16 *	0.13	0.18 *	0.03			
8	Present-Fatalistic	2.54	0.63	0.73	−0.11	0.29 **	−0.13	0.28 **	0.58 **	−0.10	0.38 **		
9	Future-Positive	3.66	0.54	0.74	0.25 **	−0.17 *	0.29 **	−0.21 **	−0.25 **	0.26 **	−0.24 **	−0.25 **	
10	Future-Negative	3.03	0.62	0.72	−0.34 **	0.39 **	−0.32 **	0.32 **	0.70 **	−0.20 **	0.13	0.52 **	−0.22 **

Note: ** *p* < 0.01. * *p* < 0.05.

## Data Availability

The data presented in this study are available from the corresponding author upon reasonable request.

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
