# Peer review of "Leaving Past Adversities Behind: Gratitude Intervention Compensates for the Undesirable Effects of Past Time Perspectives on Negative Affect"

_ijerph, 2022, doi:10.3390/ijerph191912964_

Round 1
Reviewer 1 Report
I think the present study is overall an excellent work.
So I only have some few recommendations for improving the manuscript:
(1) in the sections "1.1. Conceptual background for the current research" and /or "1.2. The present study" I a miss some explanation and description of the study by Burzynska, and Stolarski: Please provide some more details to the readers. This might me some generic passage or summary of the study in some few sentences would help a lot.
(2). please check if you can find some way to better connect the theoretical assumptions with the formulated hypothesis in the current study. I mean to add some responses to some WHY?-questions. This can be founded on some more references or on some well formulated own arguments (or both).
(3) the conducted study is on online-study which collected data via social-media. So please discuss also some possible disadvantages like selection effects, possible not to reach specific parts of the population. This may also include some arguments why these well-known disadvantages may not affect the results of the current study very much.
Author Response
I think the present study is overall an excellent work.
So I only have some few recommendations for improving the manuscript:
(1) in the sections "1.1. Conceptual background for the current research" and /or "1.2. The present study" I a miss some explanation and description of the study by Burzynska, and Stolarski: Please provide some more details to the readers. This might me some generic passage or summary of the study in some few sentences would help a lot.
Authors' response: Actually the paper mentioned by the Reviewer is a conceptual analysis article, not an empirical one. Hence, there are no results to mention. However, we have now broadened the description of the conceptual models proposed by Burzynska and Stolarski by adding the information about the two models that were not supported in the empirical study conducted by Burzynska-Tatjewska, Matthews and Stolarski (2022) - see lines 44-47 for the newly added part.
(2). please check if you can find some way to better connect the theoretical assumptions with the formulated hypothesis in the current study. I mean to add some responses to some WHY?-questions. This can be founded on some more references or on some well formulated own arguments (or both).
Authors' response: We have added a brief description of the possible mechanisms that could shape the hypothesized moderation effects. We believe that with this broadened rationale, the hypotheses are now fully justified.
(3) the conducted study is on online-study which collected data via social-media. So please discuss also some possible disadvantages like selection effects, possible not to reach specific parts of the population. This may also include some arguments why these well-known disadvantages may not affect the results of the current study very much.
Authors' response: Thank you for catching that - we do agree with that point, and we indeed omitted this limitation. We have now added the online-study-related limitations to the Limitations section (see lines 394-400).
Reviewer 2 Report
The article has a good quality.
It does bring new and highly original things.
Clarifying the objectives would add value to the study.
Could be improved by explaining the usefulness of the research.
The bibliography can be adjusted.
Author Response
The article has a good quality.
It does bring new and highly original things.
Clarifying the objectives would add value to the study.
Could be improved by explaining the usefulness of the research.
Authors' response: We assume that the two points above refer to the same issue (i.e., that the latter is a suggestion how to fix the former). We have added the practical objective of the study (i.e., providing valuable information for positive psychology practitioners) in the paragraph where we define the major aims of our research (see lines 70-75).
The bibliography can be adjusted.
Authors' response: We have double-checked the references to make them fully consistent with the journal's formatting standards.